# A Culturomics-Based Bacterial Synthetic Community for Improving Resilience towards Arsenic and Heavy Metals in the Nutraceutical Plant *Mesembryanthemum crystallinum*

**DOI:** 10.3390/ijms24087003

**Published:** 2023-04-10

**Authors:** Noris J. Flores-Duarte, Eloísa Pajuelo, Enrique Mateos-Naranjo, Salvadora Navarro-Torre, Ignacio D. Rodríguez-Llorente, Susana Redondo-Gómez, José A. Carrasco López

**Affiliations:** 1Departamento de Microbiología y Parasitología, Facultad de Farmacia, Universidad de Sevilla, c/ Profesor García González, 2, 41012 Sevilla, Spain; nflores@us.es (N.J.F.-D.); snavarro1@us.es (S.N.-T.); irodri@us.es (I.D.R.-L.); 2Departamento de Biología Vegetal y Ecología, Facultad de Biología, Universidad de Sevilla, Avda. Reina Mercedes, s/n, 41012 Sevilla, Spain; emana@us.es (E.M.-N.); susana@us.es (S.R.-G.)

**Keywords:** halophytes, ice plant, synthetic communities, biofertilizers, culturomics, heavy metals, metalloids, metabolomics

## Abstract

Plant-growth-promoting bacteria (PGPB) help plants thrive in polluted environments and increase crops yield using fewer inputs. Therefore, the design of tailored biofertilizers is of the utmost importance. The purpose of this work was to test two different bacterial synthetic communities (SynComs) from the microbiome of *Mesembryanthemum crystallinum*, a moderate halophyte with cosmetic, pharmaceutical, and nutraceutical applications. The SynComs were composed of specific metal-resistant plant-growth-promoting rhizobacteria and endophytes. In addition, the possibility of modulating the accumulation of nutraceutical substances by the synergetic effect of metal stress and inoculation with selected bacteria was tested. One of the SynComs was isolated on standard tryptone soy agar (TSA), whereas the other was isolated following a culturomics approach. For that, a culture medium based on *M. crystallinum* biomass, called Mesem Agar (MA), was elaborated. Bacteria of three compartments (rhizosphere soil, root endophytes, and shoot endophytes) were isolated on standard TSA and MA media, stablishing two independent collections. All bacteria were tested for PGP properties, secreted enzymatic activities, and resistance towards As, Cd, Cu, and Zn. The three best bacteria from each collection were selected in order to produce two different consortiums (denominated TSA- and MA-SynComs, respectively), whose effect on plant growth and physiology, metal accumulation, and metabolomics was evaluated. Both SynComs, particularly MA, improved plant growth and physiological parameters under stress by a mixture of As, Cd, Cu, and Zn. Regarding metal accumulation, the concentrations of all metals/metalloids in plant tissues were below the threshold for plant metal toxicity, indicating that this plant is able to thrive in polluted soils when assisted by metal/metalloid-resistant SynComs and could be safely used for pharmaceutical purposes. Initial metabolomics analyses depict changes in plant metabolome upon exposure to metal stress and inoculation, suggesting the possibility of modulating the concentration of high-value metabolites. In addition, the usefulness of both SynComs was tested in a crop plant, namely *Medicago sativa* (alfalfa). The results demonstrate the effectiveness of these biofertilizers in alfalfa, improving plant growth, physiology, and metal accumulation.

## 1. Introduction

Plant-growth-promoting bacteria (PGPB) are microorganisms that exhibit beneficial effects on plant health and growth, antagonize disease-causing microbes, and improve nutrient availability and assimilation [1]. These PGPB can be located either in the rhizosphere or within the tissues, living as endophytes [2]. PGPB can promote plant growth either by direct mechanisms (e.g., phytohormones production, nutrient acquisition such as N, P, and K, etc.) and indirect mechanisms (including cell-wall-degrading enzymes, production of antagonistic metabolites against pathogens, enhancement of systemic resistance, diminution of plant stress, etc.) [3].

Bacterial contribution to plant health can be studied using advanced technologies, such as metagenomics. NGS techniques allow us to know not only the composition of the entire plant microbiome, but also acquire a functional knowledge of the roles of these microbes as plant health guarantors [4]. Furthermore, the microbiome can regulate the quantity of certain metabolites within the endosphere, biotransform certain plant metabolites, and stimulate the production of other chemicals for adaptation to environmental conditions [5].

In the last decade, the field of plant biostimulants based on PGPB has gained relevance and it is considered an emerging strategy to enhance crop yield and resilience to the changing climate [5]. Because the isolation of a single PGPB with all the required properties and activities is very difficult and improbable, the most common strategy used to design a biofertilizer is based on the presence of a “core genome”, i.e., the selection of 2–4 bacterial strains whose genomes contain all the required traits. Thus, the existence of most of the desired activities is guaranteed [6,7]. These consortiums are also called synthetic microbial communities (SynComs) [8].

However, in many cases, cultivation of PGPB is hindered by the existence of specific culture media adapted to particular bacterial strains. Conventional culture media recover less than 10% of plant-associated microbiota [9]. In this sense, plant-based culturomics has emerged as an alternative to commercial media containing meat extract, peptone, etc., such as nutrient agar, TSA, etc. [9]. Culturomics is a high-throughput method used to comprehensively culture and identify strains or species in samples. It was first developed for the microbial characterization of the human gut, where cultivation and identification yielded 400 non-previously cultivated species [10]. Later on, it was applied to the plant microbiome. It is based on the optimization of culture conditions by adding specific substances of the environmental niche and specific incubation conditions. This approach was conceived as an alternative and complementary method for the metagenomic characterization of complex microbial populations [9].

*Mesembryanthemum crystallinum* (family Aizoaceae, Caryophyllales), commonly known as the ice plant, is a halophyte introduced in the Mediterranean coasts that can be consumed as an edible plant. It is characterized by the ability to switch metabolism from C3 to Crassulacean acid metabolism (CAM) under stress conditions [11]. This change is determined by an interplay between genes and environmental conditions, such as drought, salinity, or exposure to UV radiation, during *M. crystallinum* development [11]. In this context, this plant is a model halophyte that has been used to investigate physiological, biochemical, and molecular mechanisms of tolerance and resistance to abiotic stresses. Another important fact regarding this species is the existence of bladder cells on leaves and stems surfaces [12]. Bladder cells contain phytochemical compounds, such as phenolic compounds, myo-inositol, pinitol, and several minerals, with useful pharmaceutical and nutraceutical activities against human diseases. Thus, *M. crystallinum* is considered a high-value crop known to be effective against diabetes: pinitol helps control blood-sugar levels, whereas myo-inositol has a significant role in lowering the levels of lipids and cholesterol and maintaining blood pressure [13,14].

In addition, *M. crystallinum* is known for its cadmium and chromate phytoremediation capacity [15]. In this regard, several metal-resistant endophytes have been isolated from this plant in both metabolism states, C3 and CAM, which display good phytoremediation potential [16]. These bacteria include *Providencia rettgeri*, isolated during the C3 phase, and *Paenibacillus glucanolyticus* and *Rhodococcus erythropolis* isolated during the CAM phase.

In spite of the multiple applications of *M. crystallinum*, its cultivation is far from extensive due to its low germination rate and slow growth. In this regard, halotolerant rhizosphere bacteria associated with *M. crystallinum* have been found [17], indicating that microbe inoculation can help plants endure adverse conditions, such as high salinity. In addition to helping overcome abiotic stresses, PGPB can promote metabolic changes, which could positively influence the production of relevant metabolites present in the ice plant [4].

Thus, the aims of this study were as follows: (a) to select, identify, and characterize a bacterial synthetic community of metal/loid-resistant strains with the best plant-growth-promoting properties; (b) to identify metal/loid resistance traits in the genomes of the selected strains; (c) to determine the effect of the selected consortiums on plant growth and physiology, metal accumulation, and metabolomics in the ice plant in the presence of metals; and (d) to test the usefulness of these bioinoculants in a crop plant.

## 2. Results

### 2.1. Determination of the Minimal Inhibitory Concentrations (MICs) for Arsenic and Metals and Selection of Bacterial Consortiums

In general, As resistance displayed by bacteria isolated on TSA was lower than those isolated on MA (Table 1). In fact, within the TSA group, none of the bacteria reached the highest MIC for As (512 mg∙L^−1^), whereas in the MA group, 3 out of 6 bacteria displayed the highest level of resistance. Regarding the resistance towards Cd, there was a wide diversity of MIC values ranging from 0.5 to 32 mg∙L^−1^, without significant differences between the TSA and MA groups. Regarding the resistance to Cu, we found again within the MA group, a higher resistance rate, because five out of six strains had MIC of 256 mg∙L^−1^. Finally, the resistance towards Zn showed a different pattern, because the TSA collection had higher levels of resistance (with MICs between 64 and 128 mg∙L^−1^) compared to the MA group.

The following strains were finally selected: for the TSA consortium, S3 (identified as *Pantoea agglomerans*) + R5 (identified as *Bacillus velezensis*) + H4 (identified as *Pseudomonas chlororaphis*) were identified, and for the MA consortium, MS2 (identified as *Priestia megaterium*, formerly *Bacillus megaterium*) + MR4 (identified as *Bacillus subtilis*) + MH8 (identified as *Pseudomonas gessardii*) were identified. It should be noted that in every SynCom, there was a representant of different plant compartments, i.e., soil, roots, and shoots. 

### 2.2. Identification of Arsenic and Metal Resistance Traits in the Core Genome of SynComs

Traits for resistance towards metals and metalloids were identified in all strains (Table 2). *Pantoea agglomerans* S3 carries traits conferring resistance towards As (protein Yfg), several determinants for Zn resistance (such as YfgC, Znt, and ZitB), copper (in particular, the two-component system copCD for transporting Cu to the periplasmic space and binding it to CopC), and an efflux of ATPase for divalent cations (Cd, Pb, Zn, and Hg). *Bacillus velezensis* R5 has resistance determinants, such as several ATPases able to efflux Co, Cd, Hg, Pb, and Zn. Furthermore, it has determinants for copper resistance, in particular the CueR regulator belonging to the Mer family and determinants conferring resistance to As and Sb. For its part, the third component of the TSA SynCom, i.e., *Pseudomonas chlororaphis* H4, carries several genes providing resistance towards As (ArsR, Acr3, two copies of ArsH and ArsO), efflux ATPases for divalent metals, and the copper chaperone CopZ.

Regarding the MA SynCom, *Priestia megaterium* MS2 possesses genes conferring resistance towards As (AsrR, Acr3); several determinants for Zn resistance (ZnuBC, Zur, and Znt, belonging to the ZIP family) and for Cu resistance (copACDZ, CutC); the Mg/Co transport protein CorA; and several copies of metal-transporting ATPases able to translocate Pb, Cd, Zn, and Hg. *Bacillus subtilis* MR4 has resistance genes for As (Acr3, AsrR), and Cu (CueR of the MerR family and CopABCD). In addition, several copies of CzcD and metal-transporting P-type ATPases have been found. Finally, *Pseudomonas gessardi* MH8 possesses several As-resistance determinants (ArsR, Acr3, ArsH, ArsO), genes for Cu resistance (the CusSR two-component system and copABCD) for Cd resistance (CadA), and metal-transporting ATPases for the efflux of different divalent cations, such as Pb, Cd, Zn, Co, and Hg.

### 2.3. Effect of Inoculation on Plant Biomass

The plant-growth-promoting properties of the bacterial consortiums can be deduced from the increase in plant biomass in the absence of metals (Figure 1). The dry biomass of *M. crystallinum* shoots increased two- and three-fold when inoculated with TSA and MA SynComs, respectively (Figure 1A). Concerning the roots’ biomass, it is noteworthy that, despite being in contact with the bacteria, their increases were smaller than those of the shoots, particularly 15 and 50%, respectively, when inoculated with either consortium (Figure 1B).

The presence of metals had a negative effect on *M. crystallinum* biomass as is evidenced by the dry weight reduction of 85% in non-inoculated plants (Figure 1). Inoculation with the bacterial consortiums fully mitigated the deleterious effect of metals, particularly the MA SynCom, which was able to increase shoot biomass by up to 10 times and root biomass by 15-fold in the presence of metals. The TSA consortium improved both shoot and root biomass by 6-fold. Both in the absence and the presence of metals, the consortium based on the culturomics approach showed better promoting properties compared to the TSA consortium when applied to the host plant.

### 2.4. Effect of Inoculation on Plant Nutritional and Physiological Parameters

Besides diminishing plant growth, the presence of metals affected the nutritional status of the plant (Table 3). Fe content decreased by around 11% in metal-stressed plants. In contrast, the content of several other nutrients incremented between 3% (Ca) and 21% (K). On its side, the content of Na increased by 13%. Regarding P, its concentration also increased by 14%, whereas N did not show significant differences between plants grown in the absence and presence of metals.

Concerning the physiological status of the plant, the determination of the photosynthetic status (Figure 2) showed that the maximum quantum efficiency of PSII photochemistry (Fv/Fm) suffered a moderate but significant reduction (around 20%) after exposition to metals. This parameter correlates with the number of functional PSII reaction centers, so moderate reductions could suggest that this plant may be relatively tolerant towards metals. In our case, inoculation with either of the two SynComs fully restored the maximum quantum efficiency of PSII (Fv/Fm). On its side, the quantum efficiency of PSII (ΦPSII) was diminished by 40% in the presence of metals and, again, inoculation with either of the SynComs was able to fully recover the functionality of PSII.

### 2.5. Accumulation of Metals in Plant Tissues

Data on metal accumulation in shoots and roots (Table 4) lead to the conclusion that *M. crystallinum* behaved as an excluder plant because the values from all tissues were very low, particularly in shoots. The tissue distribution of metals/metalloids was differential, being the concentration of all the elements (As, Cd, Cu, and Zn) in roots higher than in the shoots. In our study, inoculation with both SynComs, and particularly with the MA consortium, led to a moderate increase in metal accumulation, which incremented between 1.2 and 3-fold depending on metal/loid and tissues. One exception was the accumulation of Zn in roots; in this case, inoculation with both TSA and MA consortiums diminished the concentration of Zn by 1.27- and 1.36-fold, respectively. Despite the increase in all metal concentrations in the shoots, all of them were far below the limits established for plant toxicity. 

### 2.6. Metabolomics Analysis

A preliminary metabolomics study was initiated in order to see the effect of exposure to metals on the metabolome of the plant. The data can be seen in Figure 3. Regarding amino acid metabolism, on one hand, three showed a significant increment in concentration in the presence of arsenic and metals, namely alanine (1.25-fold increase), glutamate (1.35-fold increase), and proline (1.3-folds increase). On the other hand, aspartate and tyrosine showed decreased concentrations in metal-treated plants (reductions of 33% and 54%, respectively) (Figure 3A). With regard to the levels of organic acids, this plant seems to accumulate high levels of citrate and malate in comparison to other organic acids (around 3 and 4 mM for citrate and malate, respectively) (Figure 3B). Moreover, the concentration of malate increased by 25% in the presence of metals. Concerning the concentration of sugars (Figure 3C), this plant seems to accumulate large quantities of glucose, (between 8–11 mM), and, to a lesser extent, fructose, whose concentrations decrease significantly in the presence of metals/metalloid (between 30 to 35% in both cases). The concentrations of *myo*-inositol and sucrose did not show significant differences upon metal exposure. In relation with other metabolites, a significant increase in chlorogenate concentration was observed (25% increase), while no significant differences in choline and trigonelline concentrations between the control and metal-exposed plants were detected (Figure 3D).

Inoculation with both consortiums (Table 5) led to higher concentrations (between 140–260% depending on the consortium and the particular compound) of organic acids, including citrate, formate, fumarate, malate, and succinate. However, the concentrations of some amino acids, such as leucine and phenylalanine, decreased to approximately half value with the inoculation of both SynComs. Additionally, there were contrasting results in the case of proline; inoculation with TSA SynCom led to a strong decrease (48%) in the concentration of this particular amino acid, while inoculation with the MA SynCom seemed not to affect it. With respect to sugars, there were also contrasting results; while the TSA consortium increased the concentration levels of fructose and glucose, the MA consortium significantly boosted those of sucrose (more than 3-fold). Inoculation with the TSA consortium also diminished (35% reduction) the accumulation of *myo*-inositol. Finally, choline accumulation was improved by inoculation with the MA consortium, whereas that of trigonelline was enhanced by inoculation with the TSA consortium.

### 2.7. Testing the Usefulness of the SynComs in a Crop Model Plant 

To test the validity of the culturomics approach in a plant other than the host, *Medicago sativa* (alfalfa) was selected. A summary of the results obtained by inoculation of *M. sativa* plants with both SynComs is shown in Table 6, while the full report is shown in Appendix A.

Three different categories were investigated, i.e., plant growth, physiological parameters, and metal accumulation. The plant-growth-promoting activities of the bacterial consortiums also had an effect on the biometric parameters of alfalfa plants, which can be inferred from the increase in plant biomass: inoculation with the TSA SynCom increased shoots and roots biomass by 19 and 16%, respectively. The MA consortium had, for its part, better growth promotion activity with gains of 23 and 41% for shoots and roots biomass, respectively.

The presence of metals had a negative effect on alfalfa biomass because the dry weight of shoots was reduced by 85% in non-inoculated plants. Inoculation with the bacterial SynComs partially mitigated the deleterious effect of metals, particularly the TSA consortium, which was able to increase shoot biomass by 2-fold and root biomass by over 4-fold. The inoculation of the MA SynCom also ameliorated shoot biomass by 2-fold and root biomass by 2.5-fold. The bigger yield in shoot biomass was related to an increase in the number and size of the leaves (Table 6).

The physiological status of the alfalfa plants in the presence of metals/metalloids was significantly improved upon inoculation, particularly with the MA consortium. This SynCom was able to protect the photosynthetic apparatus and ameliorate the maximum quantum efficiency of PSII around 5%, whereas the electron transport rate (ETR), the net photosynthesis rate (AN), and the stomatal conductance (g_s_) increased by around 2-fold (Table 6). 

Finally, inoculation with SynComs affected alfalfa metal accumulation, in both shoots and roots. The consortiums significantly increased the accumulation of most elements, with the exception of As and Cu, when the TSA consortium was inoculated (Table 6). However, the MA SynCom promoted a higher accumulation in shoots. For instance, concentrations of As and Zn were doubled and tripled, respectively, upon inoculation with the MA SynCom. In this case, it must be taken into account that the final concentration of As in shoots may surpass the thresholds for metal toxicity in plants. This could be related to the different metal resistance traits found in the members of both SynComs and with particular genetic traits controlling metalloid accumulation in alfalfa with regard to *M. crystallinum.*

## 3. Discussion

### 3.1. Deleterious Effects of Environmental Stresses on Mesembryanthemum crystallinum

*M. crystallinum*, a moderate halophyte also called the ice plant, is recently deserving of great interest due to its multiple applications, not only as an edible plant but also because of its nutraceutical properties and as a physiological model for the transition between C3 and CAM metabolisms [11,14].

Imposed abiotic stresses, such as climate change, high temperatures, salinity, and heavy metal soil pollution, have deleterious effects on the growth, yield, and physiology of all plants, particularly of the ice plant [11,12,15]. This plant is known to be capable of altering the type of metabolism, from C3 to CAM under different stress situations, such as salinity, UV radiation, metals, etc. [11,13,20].

### 3.2. Role of Metal-Resistant Biostumulants in Alleviating Plant Stress

In this context, the role of biostimulants in mitigating the effects of climate change on crop performance is being increasingly appreciated [5]. In previous works, the effectiveness of several PGPR and endophytes for improving the resilience and yield of the ice plant under some stress situations has been demonstrated. For instance, the strains *Streptomyces* sp. PR-3 and Bacillus sp. PR-6, which are tolerant to high salt concentrations and show different PGP abilities, improved the growth of the associated plant under saline conditions [17].

However, to our knowledge, a culturomics approach has not been assessed yet for the design of bacterial synthetic communities (SynComs), aimed specifically at ameliorating the performance of the plant *M. crystallinum* under metal stress conditions. To achieve this, a specific medium, called Mesem Agar (MA), was previously developed simply by triturating whole *M. crystallinum* plants in the presence of buffer and agar [18]. This strategy has allowed us to obtain two different bacterial collections, one in standard TSA medium and another one in MA. The bacteria selected in our work must display a number of PGP abilities, rhizosphere and endophytic enzymatic activities, and high metal/loid resistance. The strategy for the design of SynComs and strain selection followed the “core genome” model [6], in which the highest number of properties was present, together with the highest metal/loid resistance, within the consortium genomes. Thus, finally, the selected TSA SynCom strains were *Pantoea agglomerans* S3, *Bacillus velezensis* R5, and *Pseudomonas chlororaphis* H4, whereas in the MA SynCom, the selected strains were *Priestia megaterium* MS2, *Bacillus subtilis* MR4, and *Pseudomonas gessardii* MH8. These genera of bacteria are usually employed as PGPR, and many different strains have been reported to carry plant-growth-promoting or biocontrol activities. For instance, *Pseudomonas chlororaphis* strain ST9 promotes the growth of rice plants and displays several PGP and biocontrol activities [21]. The determinants involved in these activities are being deciphered [22]. However, they are not highly competitive in the rhizosphere, which may prevent or limit its use as a plant probiotic [21]. The fact that we isolated the strain as endophytes of shoots may solve the problem of low competitiveness in the rhizosphere because the bacterium localizes inside the plant tissues. *Bacillus velezensis* is a well-known PGPR able to ameliorate the yield of an array of plants and to secrete bioactive molecules into the rhizosphere [23,24]. On the other hand, the strain R7 of *Pantoea agglomerans* with PGP properties, such as IAA and siderophore production, P solubilization, N2 fixation, and high resistance to several metals, has been used for successful inoculation of a wide variety of winter plants in an urban orchard [25]. Other authors also proposed the use of several strains of *P. agglomerans* [26,27,28]. However, this bacterium belongs to biosafety group 2 because it can cause opportunistic infections in immunocompromised patients [29]. Because it is an endophytic bacterium, it has been reported that it can also cause infections in gardeners through punctures with colonized plants. Thus, it is not allowed to be used as a biofertilizer in Europe because only GRAS (generally recognized as safe) microorganisms can be used for plant inoculation. In order to overcome this problem, using the metabolites secreted by the bacterium instead of living microorganisms was proposed [28].

In the same way, the selected genera of bacteria for the MA SynCom are usually employed as biofertilizers. For instance, *Bacillus subtilis* is known for its PGP and biocontrol properties [30,31]. Similarly, metal-tolerant strains of *Pseudomonas gessardii* promoted plant growth, plant physiology, and Pb immobilization in sunflower [32]. Finally, *Priestia megaterium* (formerly *Bacillus megaterium*) is a bacterium with multiple applications in biotechnology, including production of polyhydroxibutyrate, production of recombinant proteins, vitamins, etc. [33], considered as a living factory [34,35]. As a PGPR, *Priestia megaterium* promotes plant growth and induces the systemic resistance mechanism in several plants, such as *Camellia sinensis* [36], rice [37], oilseed rape [38], etc.

### 3.3. Metal Resistance in the Selected Strains

In recent studies, metal-tolerant strains were isolated from *M. crystallinum* plants subjected to Cd stress. Interestingly, different isolates could be recovered from the rhizosphere of plants displaying C3 or CAM metabolisms (the latter was preinduced by salt treatment). Although two *Providencia* strains were found among the most tolerant to Cd in C3 plants, Gram-positive species such as *Paenibacillus* and *Rhodococcus* were the most tolerant species to Cd in CAM plants [16]. In our culturomics approach, both Gram-positive and Gram-negative species were also present and exhibited the highest levels of metal tolerance. In fact, all of these bacteria are known for their high resistance towards an array of metals [39,40,41]. Moreover, none of them have safety issues that impair their use as biofertilizers, because they are considered GRAS.

The metal resistance traits of these strains were codified by the presence of multiple metal-resistance determinants in their genomes. In general, several arsenic (*arsC, arsH, arsO*), copper (*copABCD, cutC, cusSR, cue*), cadmium (*cadA*), and zinc (*znt, zitB, czc*) resistance genes and operons were identified. In addition, multiple ATPases putatively involved in the efflux of several metals from bacterial cells were found.

### 3.4. Effect of Inoculation with SynComs on Growth, Physiology and Metal Accumulation by M. crystallinum

The application of both consortiums to host plants produced a clear increase in the biomass of shoots and roots, particularly the MA SynCom. Moreover, the effects were more pronounced in the presence of metals, supporting the application of these SynComs in situations of metal pollution. Analogously, metal-resistant PGPR, including *Providencia rettgeri, Paenibacillus glucanolyticus,* and *Rhodococcus erythropolis,* appear as promising robust microorganisms for biotechnological applications in phytoremediation projects using *M. crystallinum* plants [16].

Concerning the physiology of the plant, not many studies have approached this aspect in the presence of metals. However, previous reports showed that chlorophyll a fluorescence confirmed the high tolerance to Cd of the photosynthetic apparatus of *M. crystallinum* in both metabolic states, namely C3 and CAM [42]. More differences were observed for metal accumulation, as described later. Other authors reported that this plant adapts to copper or zinc by modulating water status by triggering a water-conserving strategy involving downregulation of several aquaporin genes, reduced transpiration, and an accelerated switch to CAM metabolism [20]. In fact, it is known that *M. crystallinum* is a facultative CAM plant, and it is able to change from C3 to CAM metabolism in the presence of metals. Pre-treatment with salt induces CAM metabolism and salt-adapted plants can better tolerate and accumulate higher concentrations of Cd than C3 growing plants, particularly in roots [43]. Other factors, such as LED light quality synergically with salt, affects the phytochemical production of *M. crystallinum* [44]. In our case, there was a moderate negative influence of metals on the maximum quantum efficiency of PSII. This parameter correlates with the number of functional PSII reaction centers, so moderate reductions could suggest that this plant may be relatively tolerant towards metals, in accordance with previous suggestions by other authors [15]. However, inoculation with both consortiums fully restored the functionality of the PSII. 

In our study, metal/loid pollution affected, not only their accumulation, but also the content of macro- and micronutrients in *M. crystallinum* plants. Elements such as Ca, Mg, and K suffered significant enhancements, while Fe content decreased by around 11%, probably as a consequence of its transport inhibition in order to minimize the entrance of toxic elements and/or as a consequence of Fe homeostasis disturbance [45]. For its part, Na content increased by 13%. This plant is moderately halophilic, so the accumulation of Na can be a mechanism for counterbalance metal toxicity. In fact, previous studies have demonstrated the mutual influence of NaCl and Cd in the expression of marker genes involved in heavy metal trafficking in this plant [43].

The accumulation of toxic metals in plant tissues is important not only to establish the possible application of this plant in phytoremediation [15,16], but also to establish whether its cultivation in polluted soils could be safe; particularly considering that it has pharmaceutical applications [13,14]. Our results demonstrate that the plant accumulated metals mainly in roots, with a strong limitation of translocation to the shoot. These results are consistent with previous data in the presence of Cd [15]. In this case, the authors reported that the plant accumulated Cd mainly in the roots and limited the translocation of this metal to shoots. However, the behavior of this plant, with regard to metal exposure, also depends on particular metals. In fact, the response to chromate appears to be completely different from that to cadmium, and *M. crystallinum* has been reported to accumulate Cr mainly in shoots [15]. In consequence, these authors have proposed the plant for Cd phytostabilization and Cr phytoextraction.

Inoculation with both SynComs, particularly MA, significantly increased accumulation of metal in shoots. Metal-resistant PGPR can greatly affect the mobilization of metals in soil and their accumulation in plants by mechanisms including the secretion of organic acids, siderophores, and metallophores by adsorption to bacterial exopolysaccharides, complexation, oxidation/reduction, etc. [46]. Furthermore, plants have also developed multiple mechanisms to counteract the negative effects of metals, such as limitation of metal uptake, limitation of water uptake, root lignification, complexation of metals by different substances, accumulation in “inert compartments”, such as vacuoles and cell walls, induction of antioxidant system, accumulation of secondary metabolites, etc. [47,48,49]. The possibility of modulating metal accumulation upon selected inoculation has been reported [50]. Moreover, beneficial bacteria can modulate the expression of plant genes to help them tolerate heavy metals [51]. Finally, photosynthetic metabolism C3 or CAM (the latter induced by salt application) affects the stress response to metals (antioxidant machinery, proline accumulation, etc.) and metal accumulation [43].

The overall results of metal accumulation showed that the values were always below that of the metal toxicity thresholds in plants. These thresholds are 5–20 mg∙Kg^−1^ for As; 5–30 mg∙Kg^−1^ for Cd; 2–20 mg∙Kg^−1^ for Cu; and 100–400 mg∙Kg^−1^ of Zn [20]. The results suggest that this plant is able to thrive on polluted soils when assisted by metal-resistant SynComs of rhizosphere and endophyte microorganisms, and it could be safely used for pharmaceutical or nutraceutical purposes. 

### 3.5. Improving the Accumulation of Desired Metabolites by Synergy Application of Metal Stress and Inoculation

Only few studies have been conducted on the metabolomics of this plant [4,52], despite its interest in pharmaceutical and nutraceutical applications. Some members of the *Mesembryantheaceae* family, such as *Mesembryanthemum cordifolium,* accumulate nine alkaloids with antidepressant activity, in particular the compound called mesembrane, in roots [53]. Furthermore, *M. crystallinum* has antioxidant, antihypertensive, hypoglycemic, and nootropic activity due to its polyphenolic compounds [14]. As part of the plant response towards metals, *M. crystallinum* is able to adapt its metabolome. Proline is an amino acid typically accumulated under a wide variety of stress conditions, being in fact, a marker of abiotic stress [54]. Besides its role as an osmolyte when plants are exposed to salt, proline plays additional roles during stress, such as metal chelator, antioxidant, and signaling molecule [55]. Many examples of an increase in proline levels under metal stress have been reported, for instance, wheat or tobacco [56,57]. In our case, proline content enhancement of 1.3-fold was observed, according to previous data in this plant [20]. Other studies reported an increase in proline upon induction due to diverse stresses, such as nitrogen starvation [58], salt [4], and LED light quality [44]. Another amino acid induced in our study was glutamate. Glutamate can act as a metal chelator. In fact, exogenous application of glutamate was used for Cd and Cr phytoremediation [59,60]. In particular, *M. crystallinum* can accumulate this and other amino acids for metal chelation, although the chelation capacity of amino acids is much lower than that of organic acids [61]. In addition, glutamate is one of the glutathione building blocks, together with cysteine and glycine. The accumulation of glutathione (and phytochelatins) in the presence of metals is widely documented, so glutamate synthesis must be required to fulfil the synthesis of glutathione and phytochelatins [62]. With regard to organic acids, citrate and malate are known metal ligands in plants [61,63]. Moreover, these particular organic acids have been reported to increase under salt stress in *M. crystallinum* [5]. The level of sugars, such as fructose and glucose, decreased in the presence of metals. This could reflect the negative effect of metal/loid on the photosynthetic apparatus in non-inoculated plants, ultimately leading to less glucose synthesis, despite the fact that this plant is relatively tolerant to metals [42]. Concerning other metabolites, the enhancement of the levels of chlorogenate in our study is significant. Chlorogenic acid is a phenolic compound whose concentration has been reported to experiment the highest increase in the presence of metals in maize [64].

It seems clear that the metabolomic profile of *M. crystallinum* is affected by heavy metals, but from our results, it can be also concluded that this profile can be further modulated by inoculation with appropriate SynComs. In fact, inoculation with PGPR and endophytes led to changes in the metabolomic profile of the roots of *M. crystallinum* [4,17]. For instance, inoculation with *Microbacterium* spp. resulted in increased contents of metabolites related to the tri-carboxylic acid cycle and photosynthesis [4]. In our case, the most relevant changes were produced at the level of organic acids, being clearly incremented, according to previous data [65]. By contrast, amino acids such as Leu and Phe showed decreased concentrations (around 50%). Tyr also showed a marked, but not significant reduction. Phe and Tyr are amino acids belonging to the shikimic acid pathway. This pathway provides carbon skeletons not only to the aromatic amino acids Tyr, Phe, and Trp, but also to a myriad of secondary metabolites, including alkaloids, flavonoids, lignins, etc. Many of these compounds are involved in defense mechanisms against biotic and abiotic stresses and their synthesis is highly induced by them [66]. It is believed that more than 20% carbon fixed by photosynthesis can be derived to this pathway [66]. Thus, it can be possible that derivation of the synthesis to secondary metabolites needed to counteract the stress by metals can diminish the concentration of Phe and Tyr by competition.

Finally, there are some metabolites which are differently regulated upon inoculation with both SynComs. For instance, TSA SynCom decreased the levels of Pro, probably reflecting a diminution of plant stress. In addition, it increased the concentration of fructose and glucose, in agreement with improved photosynthesis rates after inoculation. Additionally, the level of *myo*-inositol, a compound reported as a plant protectant under several stress situations, such as salt and oxidative stress [67,68], was also diminished, which probably reflects a lower oxidative status in plants inoculated with the TSA consortium. The MA SynCom increased the levels of sucrose by three-fold, reflecting a higher carbon fixation. The accumulation of choline (around a 50% increase in the presence of metals), among other metabolites, upon inoculation with this SynCom is very interesting. Choline is involved protection against salt stress in halophytes [69] and other plants, such as spinach [70], because this compound is a precursor of glycine-betaine, which has been related to heavy metal tolerance in plants [71]. Choline is considered a very necessary but often forgotten essential nutrient [72], needed for the proper function of skeletal muscle and to prevent neurological diseases [73,74]. On the other hand, chlorogenic acid did not show increased levels upon inoculation, but was enhanced by the presence of metals (Figure 3D). This phenolic compound has been reported to have valuable antioxidant and antimicrobial properties [75]. Trigonelline concentration also significantly increased upon inoculation with the TSA SynCom. Trigonelline is an alkaloid derived from vitamin B5 (niacin), which is very abundant in coffee plants [76] and has important pharmaceutical applications, such as hypoglycemic, hypolipidemic, neuroprotective, antimigraine, and sedative properties [77]. Thus, our results prove the usefulness of the synergy between metal stress and SynComs in modulating the concentrations of metabolites with important pharmaceutical and nutraceutical properties. 

### 3.6. Usefulness of the Selected Strains in a Crop Plant

In spite of the fact that the culturomics approach has been designed for *M. crystallinum*, we wanted to test our SynComs in a model plant. Alfalfa (*Medicago sativa*) is a leguminous plant widely cultivated around the world, which is used mainly as fodder. From researchers’ point of view, alfalfa is considered a crop model plant because *Medicago truncatula*, another species of the same genus, is a model legume whose genome is fully sequenced and many other genetic tools (mutants, small secreted peptides (SSPs) database, transporter protein database, gene expression atlas, proteomic atlas, and metabolite atlas) are available [78]. The results indicate that this plant is very sensitive to metal pollution because growth and physiology were significantly affected [79,80]. In this regard, our results demonstrate the usefulness of the SynComs in alfalfa plants. However, when considering the data generated in this study, it can be said that the MA consortium seems to work better for the host plant, whereas alfalfa seems to perform better with the TSA consortium. In this sense, the application of a culturomics strategy guarantees the specificity with regard to the host plant. However, the selected organisms and others from the collections with PGP properties can also be used in crop plants [25,81]. Moreover, it could be possible to further modulate the accumulation of desired metabolites with pharmaceutical and nutraceutical interest by exploring other stress conditions, such as high salinity, drought, high temperature, etc. in synergy with appropriate SynCom inoculation.

## 4. Materials and Methods

In a previous work, rhizosphere bacteria and endophytes were isolated from the rhizosphere and tissues of *Mesembryanthemum crystallinum* collected at the Guadiana River mouth [18]. For the isolation of these bacteria, a culturomics strategy was developed. Thus, two different culture media were used for the isolation, namely a standard TSA medium and a medium called Mesem Agar, made from tissues of the same plant. The medium was prepared by triturating whole *M. crystallinum* plants in the presence of a phosphate buffer and adding agar to the plant extract [18]. The extracts of rhizosphere soil, roots, and shoots were streaked in the two media. All morphologically different bacterial colonies from three compartments of *Mesembryanthemum crystallinum* plants, namely rhizosphere soil, endophytes of roots, and endophytes of shoots were isolated. Two collections were obtained: one with 47 bacteria isolated on TSA and another with 33 bacteria isolated on MA. The whole collection of bacteria (80 strains in total) was prospected for plant-growth-promoting properties and secreted enzymatic activities. Thirteen characteristics, i.e., six PGP properties (phosphate and potassium solubilization, nitrogen fixation, siderophores and auxin production, and biofilm formation) and seven enzymatic activities related to penetration into plant tissues and the recycling of organic matter in the rhizosphere (namely amylase, DNAase, protease, lipase, cellulase, pectinase, and chitinase) were determined for all isolates (see Appendix A). Twelve bacteria with the highest number of properties (six isolated on TSA and six isolated on MSA) were initially selected and their resistance to heavy metals was determined as indicated below, in order to select the final SynComs with the highest possible number of traits related to metal/loid resistance and PGP properties.

### 4.1. Determination of Metal/Loid Resistance in Bacteria

The determination of the minimal inhibitory concentration (MIC) for metals was performed on a 96-well microtiter plate assay. Wells were filled with 200 µL of Müeller–Hinton broth (MHB) containing serial dilutions (base two logarithmic dilutions) of CdCl_2_, CuSO_4_, NaAsO_2_, or ZnCl_2_ (from 512 mg∙L^−1^ to 0.5 mg∙L^−1^) according to CLSI directions [82]. As a control, the last column was filled with only MHB. Overnight cultures of each bacterial strain were diluted with sterile MHB to adjust OD_600nm_ to 1.0, and the assay wells of the microtiter plates were inoculated with 5 µL from the normalized dilution. The first row of each plate was inoculated with 5 µL of sterile MHB (non-inoculated control). Plates were incubated for 24 h at 28 °C. After this period, plates were visualized, and the minimal inhibitory concentration was the lowest concentration that inhibited bacterial growth (no visual observation of turbidity).

### 4.2. Identification of Selected Bacteria by 16SrRNA Sequencing

For DNA extraction, bacterial cultures were grown in a TSB medium overnight at 28 °C and 200 rpm. One mL of culture was pelleted at maximum speed for 3 min and the pellet was resuspended in 100 µL of a buffer solution pH 8.0 containing, per mL, 6 mg Tris; 50 mg sucrose; 7.4 mg EDTA; 5 µL Triton X-100; 40 mg ammonium chloride; 30 µL lysozyme (10 mg∙mL^−1^); and 60 µL proteinase K (10 mg∙mL^−1^). The bacterial suspensions were incubated at 37 °C for 15 min, followed by 5 min at 65 °C. Then, samples were centrifuged at maximum speed for 10 min. DNA in the supernatant was precipitated by addition of 150 µL of isopropanol and centrifugation at 10,000 rpm for 10 min. DNA pellets were washed with 200 µL of 75% ethanol, dried for 5 min at room temperature, and finally resuspended in 50 µL of TE (10 mM Tris-HCl pH 8.0 containing 1 mM EDTA). The quantity and quality of DNA was checked by gel electrophoresis (0.8% agarose, 100 V, 45 min). Total gDNA was stored at −20 °C until use.

For amplification of 16S rRNA, gDNA was diluted at a ratio of 1:20 and PCR was performed with universal primers 16S 27F and 16S 1488R [83]. PCR was performed using the PCR master mix Q5 High-Fidelity DNA Polymerase (New England Biolabs) according to the instructions of the provider. Ten µL of PCR fragments (approximately at 20 ng∙µL^−1^) were mixed with 3 µL of 10 µM forward or reverse primers and sent for Sanger sequencing. To identify the microbial species, sequences were analyzed using BLAST [84] and EzTaxon [85] and by comparison with sequence databases.

### 4.3. Whole-Genome Sequencing of the Selected Strains and Identification of Metal Resistance Traits

DNA samples were sent for whole-genome sequencing at the company StabVida (Portugal) using the platforms Illumina^®^ HiSeq^®^2500/Illumina MiSeq^®^. The quality of data was studied by mapping the reads with BWA MeM [86]. The identification of the closest available reference was performed using Kraken [87]. De novo assembly of the genome was performed using SPAdes [88], and genome annotation was performed with Prokka [89]. The genes corresponding to metal resistance, in particular to As, Cd, Cu, and Zn, were prospected.

### 4.4. Preparation of the Bacterial Consortium for Plant Inoculation

In order to exclude the possibility of antagonism within the bacterial SynComs, the different strains were grown together on the same TSA plate. Each bacterium was cross stricken out with every other selected strain, making sure all of them made contact with each other. They also were grown in combinations of two on the same plate. Plates were incubated for 24 h at 28 °C and growth of every species up to the borderline with each other was taken as an indication of no antagonism.

For the preparation of the inoculant, individual cultures of each of the three strains were cultivated in 200 mL of medium Mesem Broth and incubated for 48 h at 28 °C at 200 rpm. After 48 h, the cultures were centrifuged at 8000 rpm for 5 min. The supernatants were discarded, and the pellets were resuspended in 50 mL of saline solution (0.9% w). Washing was repeated once again. Finally, 50 mL of saline solution were used to subsequently resuspend the three bacterial pellets together to conform the final inoculum. Final bacterial density of the consortiums was between 10^8^–10^9^ UCF∙mL^−1^ once diluted in water for plant inoculation.

### 4.5. Cultivation of Plants in the Greenhouse

Seeds of *Mesembryanthemum crystallinum* (collected at the Guadiamar riverbed) and commercial seeds of *Medicago sativa* (alfalfa) cv. Aragon were surface disinfected by immersing them in 70% (*v*/*v*) ethanol for 2 min, followed by immersion in commercial bleach solution (1:2, *v*/*v*) for 10 min, before they were finally washed three times with sterile, distilled water [90]. Seeds were pregerminated in water agar (9 g∙L^−1^) for 48 h at 25 °C in the dark.

Germinated seedlings in the same developmental stage were transferred to individual pots (0.5 L volume) containing an autoclaved mix of substrate, perlite, and sand (8:1:1). Pots were placed on separated trays for metal treatments and to avoid cross contamination upon inoculation. Trays were placed in a greenhouse with illumination and temperature control, under a 14 h light: 10 h dark and 26 °C: 18 °C photoperiod. Routinely, pots were watered once a week with 50 mL water per pot, except for the occasions of inoculation and metal treatments as described below. Six different treatments were established at the beginning of the experiment for two different plants species, i.e., *Mesembryanthemum crystallinum* and *Medicago sativa*. With regards to the inoculation treatments, three conditions were tested, i.e., non-inoculated, inoculated with TSA-SynCom, and inoculated with MA-SynCom. The inoculation treatments were done during planting and once a month. To do so, 50 mL of the bacterial SynCom (prepared as in Section 2.5) were mixed with 3 L water and 50 mL of this bacterial suspension was added to each individual pot. Regarding metal treatments, two conditions were used, i.e., absence of metals and presence of a mixture of metals/metalloid (1 µM Cd Cl_2_ + 20 µM NaAsO_2_ + 50 µM CuSO_4_ + 200 µM ZnCl_2_). Individual pots were watered with 50 mL of the solution containing the mix of elements. The addition of metals/metalloid was repeated at the beginning of the experiment and once a month, coincident with the inoculation treatments. The rest of the time, plants were watered with tap water. *Medicago* plants were cultivated for 2.5 months, whereas *M. crystallinum* (whose life cycle was longer) were cultivated for 5 months. Every week, trays with plants were randomly distributed on the tables of the greenhouse to avoid position effects. At the end of the experiment, the physiological state of the plants was evaluated and then they were collected for biomass, metal/loid accumulation measurements, and metabolomic analyses.

### 4.6. Determination of the Physiological Status of the Plants

Chlorophyll fluorescence was determined using a portable modulated fluorimeter (FMS-2, Hansatech Instrument Ltd., England) in fully expanded leaves (*n* = 7). Measurements were performed at midday (1600 μmol m^−2^ s^−1^). First, plants were pre-adapted to the dark for 30 min using leaf-clips. The minimal fluorescence level in the dark-adapted state (F0) was measured using a modulated pulse too small to induce significant physiological changes in the plant (<0.05 μmol m^−2^ s^−1^ for 1.8 μs). The stored data were an average taken over a 1.6 s period. Maximal fluorescence (Fm) was measured after applying a saturating actinic light pulse of 15,000 μmol m^−2^ s^−1^ for 0.7 s. The value of Fm was recorded as the highest average of two consecutive points. Values of the variable fluorescence (Fv = Fm − F0) and maximum quantum efficiency of PSII photochemistry (Fv/Fm) were calculated from F0 and Fm. The ratio of variable to maximal fluorescence correlates with the number of functional PSII reaction centers, and dark-adapted values of Fv/Fm can be used to quantify photoinhibition [91]. Light-adapted parameters were determined in the same leaf section. To do so, plants were adapted to ambient light conditions for 30 min and steady-state fluorescence yield (Fs) was recorded. The maximum fluorescence yield (Fm′) was determined by temporarily inhibiting PSII photochemistry after applying a saturating actinic light pulse of 15,000 μmol m^−2^ s^−1^ for 0.7 s. Using fluorescence parameters determined in both light- and dark-adapted states, the quantum efficiency of PSII was calculated using the expression ΦPSII = (Fm′− Fs)/Fm′.

### 4.7. Determination of Plant Growth Parameters

At the end of the experiment, plants were harvested and separated into shoots and roots. The length of roots and shoots was measured, and the biomass of roots and shoots was determined (*n* =10 per treatment). Afterwards, half of the biomass of shoots and roots was dried at 80 °C for 48 h for dry mass determination. The rest of the biomass was also dried and used for metal/loid content determination and for metabolomic analyses. A small portion of fresh roots (three pieces of 0.5 cm) was reserved for observation of root colonization by bacteria.

### 4.8. Determination of Macronutrients and Metal/Metaloid Accumulation in Plants

Portions of 0.5 g of dry mass of shoots or roots samples (a homogeneous mixture of pieces from 10 plants for treatment) were digested with 6 mL HNO_3_, 0.5 mL HF, and 1 mL H_2_O_2_ at 130 °C for 5 h. The concentrations of metals/metalloid applied in the treatments (As, Cu, Cd, Zn), together with macroelements Na, K, Mg, Ca, P, and Fe were measured by inductively coupled plasma optic spectroscopy (ICP-OES) (ARL-Fisons 3410, Verona, WI, USA) using reference materials from Fisons. Total N was determined with an elemental analyzer (Leco CHNS-932, St. Joseph, MI, USA).

### 4.9. Metabolomic Analyses

Fresh shoots were quickly frozen in liquid nitrogen and then lyophilized using a freeze dryer Lyoquest -85 plus (Telstar). Lyophilized samples (50 mg for each condition, in triplicate) were sent for metabolomic analyses to the Center for Edaphology and Applied Biology of Segura CEBAS-CSIC (Murcia, Spain). An untargeted analytical approach was followed for the determination of the concentrations of amino acids, organic acids, sugars, and other metabolites by ultra-high performance liquid chromatography coupled to a quadrupole time-of-flight mass spectrometer (UPLC-QToF-MS/MS) [92].

### 4.10. Statistical Analysis

Statistical analysis was performed with the program Statistica v. 6.0 (Statsoft Inc.). Comparison between means was done by using one-way ANOVA (F-test). Previously, data were checked for normality using the Kolmogorov–Smirnov test and for homogeneity of variance with the Brown–Forsythe test. Significant results were followed by Tukey’s tests for identification of important contrasts. In the tables and figures, significant differences at *p* < 0.05 are shown by asterisks.

## 5. Conclusions

The members of the Synthetic Bacterial Communities proposed in this study show relevant PGP properties and enzymatic activities, in addition to high levels of heavy metal/loid resistance. Both SynComs have shown beneficial effects on the host plant, i.e., they promote plant growth and improve plant physiology and the accumulation of several macronutrients. As a mechanism of adaptation towards metal stress, *M. crystallinum* accumulates proline, glutamate, and organic acids, such as citrate and malate. Inoculation with SynComs allowed modulation of the metabolome and accumulation of desired metabolites with nutraceutical properties, such as choline and trigonelline. Furthermore, consortiums increase metal content in both shoots and roots. However, the concentration of metals remains below the threshold for metal toxicity in plants, indicating that the plant is able to thrive on polluted soils when assisted by metal-resistant SynComs, and it could be safely used for pharmaceutical and/or nutraceutical purposes. Finally, our results have demonstrated the usefulness of SynComs when inoculating alfalfa plants, particularly with the TSA consortium. On the other hand, the MA consortium has better effects on the host plant, suggesting the specificity of the culturomics strategy. In summary, our data support the usefulness and potential of the culturomics approach to isolate specific beneficial microorganisms that stablish a close relationship with plants by developing tailored culture media based on their biomass. Future studies are in progress in order to determine whether other abiotic stress situations, together with inoculation with appropriate SynComs derived from the culturomics approach, may boost the accumulation of metabolites with pharmaceutical/nutraceutical purposes.

## Figures and Tables

**Figure 1 ijms-24-07003-f001:**
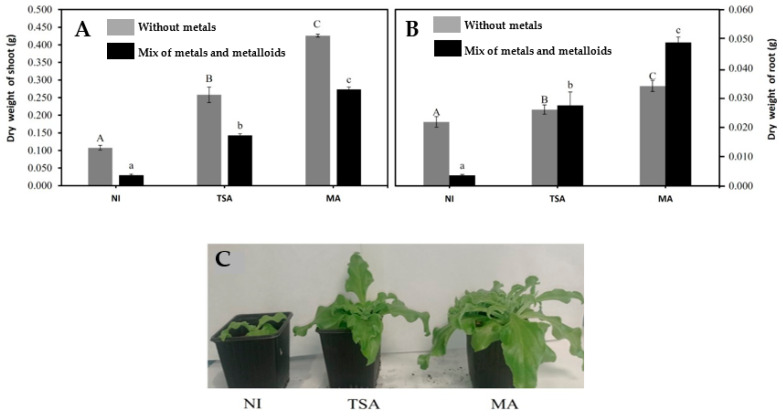
Dry biomass of shoots (**A**) and roots (**B**) of *Mesembryanthemum crystallinum* plants grown in the absence (grey bars) and presence (black bars) of a mixture of metals and metalloids (10 µM As + 1 µM Cd + 20 µM Cu + 50 µM Zn), non-inoculated (NI) or inoculated with TSA or MA SynComs. Data are means of 10 independent determinations ± standard deviations. Statistically significant differences with respect to the non-inoculated control plants at *p* < 0.05 are indicated by different letters (upper case for the plants grown in the absence of metals and lower case for plants grown in the presence of a mix of metals). (**C**) Aspect of inoculated vs. non-inoculated plants grown in the presence of metals.

**Figure 2 ijms-24-07003-f002:**
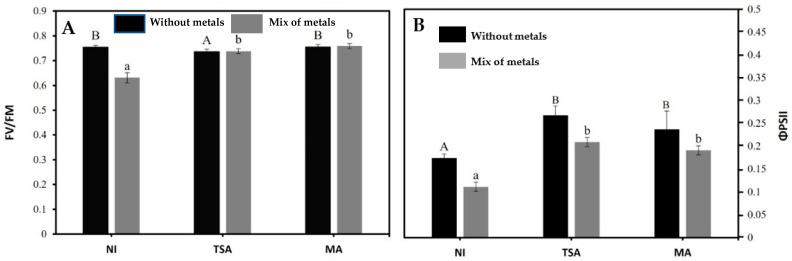
(**A**) Maximum quantum efficiency of PSII photochemistry (Fv/Fm), and (**B**). Quantum efficiency of PSII (ΦPSII) of *Mesembryanthemum crystallinum* plants grown in the absence (black bars) and presence (gray bars) of a mixture of metals and metalloids (10 µM As + 1 µM Cd + 20 µM Cu + 50 µM Zn) and inoculated with TSA or MA consortiums. NI: non-inoculated control. Data are means of 7 independent determinations in 7 different plants ± standard deviations. Statistically significant differences with regard to the non-inoculated control plants at *p* < 0.05 are indicated by different letters (upper case for the plants grown in the absence of metals and lower case for plants grown in the presence of a mix of metals).

**Figure 3 ijms-24-07003-f003:**
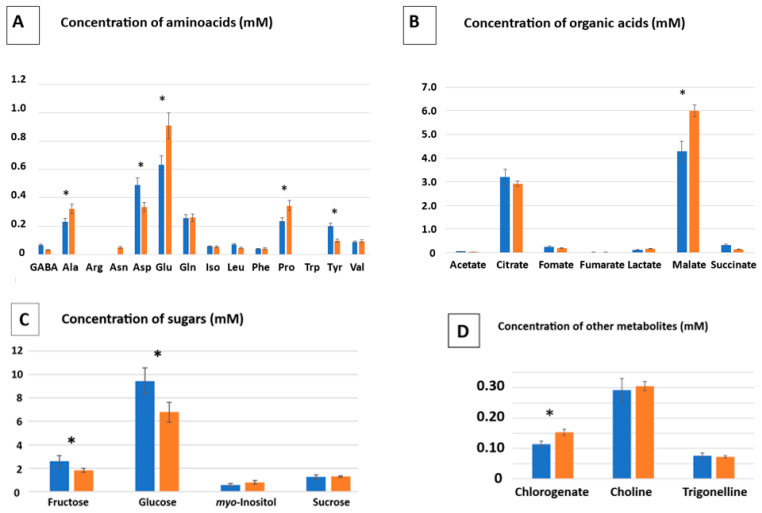
Concentration of amino acids (**A**), organic acids (**B**), sugars (**C**), and other metabolites (**D**) in the leaves of *Mesembryanthemum crystallinum* in the absence of metals (blue bars) and in the presence of a mixture of arsenic and metals (10 µM As + 1 µM Cd + 20 µM Cu + 50 µM Zn) (orange bars). Data are means ± standard deviations of three independent determinations. Statistically significant differences with regard to the control plants in the absence of metals are indicated by asterisks.

**Table 1 ijms-24-07003-t001:** Minimal inhibitory concentration (MIC) of metals/metalloids for the selected strains with the best plant-growth-promoting abilities and enzymatic activities. Strain selection was based on their PGP properties, their metal/loid resistance, and the “core genome” concept, by which most of the required PGP and metal resistance traits were present within the SynCom constituent’s genomes.

Strain	PGP Properties and Enzymatic Activities ^(1)^	MIC (As) mg∙L^−1^	MIC (Cd) mg∙L^−1^	MIC (Cu) mg∙L^−1^	MIC (Zn) mg∙L^−1^
TSA medium					
S3	Bio, N_2_-fix, Aux, Cel	128	32	256	128
R5	N_2_-fix, Aux, Pect, Cel, Amy, DNAase, Prot	128	2	128	64
H4	Bio, N_2_-fix, Amy	16	16	256	64
MA medium					
MS2	Sid, Aux, Prot, DNAase, Amy, Pect	512	2	128	32
MR4	Bio, N_2_-fix, Prot, Cel, Amy, Pect, DNAase	128	0.5	256	128
MH8	KS, PS, Sid, Bio, Pect, Amy	32	32	256	64

^(1)^ PS, phosphate solubilization; KS, potassium solubilization; N_2_-fix, nitrogen fixation; Bio, biofilm formation; Sid, production of siderophores; Aux, production of auxins; Cel, cellulose activity; Pect, pectinase activity; Amy, amylase activity; DNAase, DNAase activity; Prot, protease activity; Lip, lipase activity; Chi, chitinase activity (data published in [18]).

**Table 2 ijms-24-07003-t002:** Metals and metalloids resistance traits identified in the sequenced whole genomes of the strains selected for the design of the TSA and MA SynComs.

Strain	Metal Resistance Traits	Resistance
S3	Uncharacterized protein Yfg	As
	Exported zinc metalloprotease YfgC precursor	Zn
	Zinc transport protein ZntB	Zn
	Zinc transporter ZitB	Zn
	Copper resistance proteins CopCD	Co
	Lead, cadmium, zinc, and mercury-transporting ATPase (2 copies)	Pb, Cd, Zn, Hg
R5	Cobalt/zinc/cadmium-resistance protein CzcD	Co, Zn
	Putative metallopeptidase (Zinc) SprT family	Zn
	Cadmium, zinc, and cobalt-transporting P-type ATPase	Cd, Zn, Co
	Lead, cadmium, zinc, and mercury-transporting ATPase (4 copies)	Pb, Cd, Zn, Hg
	Copper resistance transcriptional regulator CueR (MerR family)	Cu
	Arsenite/antimonite:H+ antiporter ArsB	As, Sb
	Arsenical-resistance operon repressor (2 copies)	As
H4	Cadmium-translocating P-type ATPase	Pb, Cd, Zn, Hg
	Arsenical-resistance operon repressor	As
	Arsenical-resistance protein Acr3	As
	Arsenic-resistance protein ArsH (2 copies)	As
	Copper(I) chaperone CopZ	Cu
	Lead, cadmium, zinc, and mercury-transporting ATPase (2 copies)	Pb, Cd, Zn, Hg
	Copper-sensing two-component system response regulator CpxRCusSR	Cu
	Cobalt-zinc-cadmium-resistance protein CzcD (2 copies)	Co, Zn, Cd
	Copper-resistance operon CopABCD, 1 extra copy of CopA	Cu
	Flavin-dependent monooxygenase ArsO	As
MS2	Arsenical-resistance operon repressor (5 copies)	As
	Arsenical-resistance protein Acr3	As
	Zinc ABC transporter operon ZnuBC	Zn
	Zinc uptake regulation protein Zur	Zn
	Lead, cadmium, zinc, and mercury-transporting ATPase (4 copies)	Pb, Cd, Zn, Hg
	Zinc transporter, ZIP family	Zn
	Cobalt-zinc-cadmium-resistance protein CzcD (3 copies)	Co, Zn, Cd
	Cytoplasmic copper homeostasis protein CutC (2 copies)	Cu
	Copper-resistance operon CopACDZ (1 extra copy of copZ)	Cu
	Response regulator of zinc sigma-54-dependent two-component system	Zn
	Magnesium and cobalt transport protein CorA	Mg, Co
MR4	Arsenical-resistance protein Acr3	As
	Arsenical-resistance operon repressor	As
	Cobalt/zinc/cadmium-resistance protein CzcD (4 copies)	Co, Zn, Cd
	Cadmium, zinc, and cobalt-transporting P-type ATPase	Cd, Zn, Co
	Lead, cadmium, zinc, and mercury-transporting ATPase (2 copies)	Pb, Cd, Zn, Hg
	Copper-resistance transcriptional regulator CueR (MerR family)	Cu
	Copper-resistance proteins CopABCD	Cu
MH8	Copper tolerance protein	Cu
	Copper-sensing two-component system response regulator CusSR	Cu
	Cadmium-translocating P-type ATPase	Cd
	Arsenical-resistance operon repressor	As
	Arsenical-resistance protein Acr3	As
	Arsenic-resistance protein ArsH (2 copies)	As
	Copper-resistance operon copABCD	Cu
	Cobalt/zinc/cadmium-resistance protein CzcD (2 copies)	Co, Zn, Cd
	Lead, cadmium, zinc, and mercury-transporting ATPase (2 copies)	Pb, Cd, Zn, Hg
	Flavin-dependent monooxygenase ArsO	As

**Table 3 ijms-24-07003-t003:** Content of macro- and micronutrients in plants of *Mesembryanthemum crystallinum* in the presence of a mix of metals and metalloids (10 µM As + 1 µM Cd + 20 µM Cu + 50 µM Zn) with regard to control plants.

Culture Conditions	P Content (%)	N Content (%)	Ca Content (%)	Mg Content (%)	K Content (%)	Na Content (%)	Fe Content (mg∙Kg^−1^)
Without metals (control)	1.47 ± 0.03	0.88 ± 0.30	1.61 ± 0.02	0.73 ± 0.01	7.40 ± 0.20	2.30 ± 0.07	75.09 ± 2.74
In the presence of metal(loid)s	1.68 ± 0.01 **	0.91 ± 0.00	1.67 ± 0.00 *	0.81 ± 0.01 **	9.00 ± 0.10 *	2.60 ± 0.05 *	66.60 ± 0.97 **

One asterisk indicates significant differences with regard to the control plants in the absence of metals at *p* < 0.05; Two asterisks indicate significant differences at *p* < 0.01.

**Table 4 ijms-24-07003-t004:** Accumulation of arsenic and heavy metals in shoots and roots of *M. crystallinum* plants grown in the presence of a mixture of 10 µM As + 1 µM Cd + 20 µM Cu + 50 µM Zn, inoculated with TSA or MA SynComs. Data are means ± standard deviations of three independent determinations. Statistically significant differences with regard to the control plants in the absence of metals are indicated by asterisks. ND: non-detected.

Plant Tissue	Inoculation Treatment	As (mg∙Kg ^−1^)	Cd (mg∙Kg ^−1^)	Cu (mg∙Kg ^−1^)	Zn (mg∙Kg ^−1^)
SHOOTS	Non inoculated	ND	ND	2.08 ± 0.05	3.07 ± 0.04
	TSA	ND	ND	1.23 ± 0.07 *	5.15 ± 0.24 *
	MA	ND	ND	3.12 ± 0.16	10.92 ± 0.12 *
ROOTS	Non inoculated	ND	0.71 ± 0.00	6.89 ± 0.10	134.07 ± 0
	TSA	4.09 ± 0.27 *	1.08 ± 0.12 *	8.91 ± 0.48	98.36 ± 3.70 *
	MA	2.70 ± 0.14 *	1.17 ± 0.02 *	8.35 ± 0.38	104.99 ± 1.30 *
^(1)^ Thresholds for plant leaf tissue toxicity limits (mg∙Kg^−1^)		5–20	5–30	2–20	100–400

^(1)^ Thresholds for metal accumulation in shoots [19].

**Table 5 ijms-24-07003-t005:** Effect of inoculation with TSA or MA SynComs on the concentration of metabolites of *M. crystallinum* plants cultivated in the presence of a mixture of metals/metalloids (10 µM As + 1 µM Cd + 20 µM Cu + 50 µM Zn). Data represent the percentages with regard to non-inoculated plants, considered as 100%. Statistically significant differences with regard to the non-inoculated controls are indicated by asterisks.

	TSA SynCOM	MA SynCom
**Aminoacids**		
GABA	--	84
Alanine	87	89
Asparagine	104	---
Aspartate	100	102
Glutamate	80	75
Glutamine	123	100
Isoleucine	82	88
Leucine	43 (*)	56 (*)
Phenylalanine	56 (*)	54 (*)
Proline	48 (*)	92
Tyrosine	71	82
Valine	80	90
**Organic acids**		
Acetate	161 (*)	105
Citrate	163 (*)	251 (*)
Formate	165 (*)	152 (*)
Fumarate	187 (*)	155 (*)
Lactate	72	100
Malate	141 (*)	138 (*)
Succinate	260 (*)	153 (*)
**Sugars**		
Fructose	203 (*)	141
Glucose	175 (*)	138
myo-inostol	35 (*)	106
Sucrose	112	328 (*)
**Other metabolites**		
Choline	136	155 (*)
Trigonelline	153 (*)	125

**Table 6 ijms-24-07003-t006:** Summary of results of inoculation of alfalfa (*Medicago sativa*) plants with consortiums TSA and MA in the absence and in the presence of metals/metalloids (10 µM As + 1 µM Cd + 20 µM Cu + 50 µM Zn). Effects on plant biometric parameters, physiological status of the plants and metal accumulation. Detailed information is provided in Appendix A. Data are means ± standard deviations of three independent determinations and those significantly different from the non-inoculated control are indicated by different letters.

Metals	Parameter	Non-Inoculated	TSA	MA
Absence of metals	Shoot dry matter (g)	0.105 ± 0.10 ^(a)^	0.125 ± 0.03 ^(b)^	0.130 ± 0.09 ^(c)^
	Root dry matter (g)	0.06 ± 0.006 ^(a)^	0.07± 0.006 ^(a,b)^	0.085± 0.006 ^(b)^
	Size of leaves (cm)	0.67 ± 0 ^(a)^	0.82 ± 0.11 ^(b)^	0.79 ± 0.02 ^(b)^
	Number of leaves	12 ± 0 ^(a)^	15 ± 0.5 ^(b)^	18 ± 0 ^(c)^
	Maximum quantum efficiency of PSII	0.75 ± 0.01 ^(a)^	0.76± 0.02 ^(a)^	0.72± 0.02 ^(a)^
	Quantum efficiency of PSII	0.34 ± 0.03 ^(a)^	0.37 ± 0.04 ^(a)^	0.35 ± 0.04 ^(a)^
	Net photosynthesis rate (AN) (µM m^−2^ s^−1^)	12.32 ± 1.10 ^(a)^	13.77 ± 0.80 ^(a)^	13.18 ± 1.90 ^(a)^
	Electron transport rate	74 ± 1 ^(a)^	115 ± 14 ^(b)^	102 ± 18 ^(b)^
	Stomatal conductance (gs)(mM m^−2^ s^−1^)	390 ± 37 ^(a)^	320 ± 30 ^(b)^	315 ± 20 ^(b)^
Presence of a mix of metals: 10 µM As + 1 µM Cd + 20 µM Cu + 50 µM Zn	Shoot dry matter (g)	0.016 ± 0.004 ^(a)^	0.033 ± 0.003 ^(b)^	0.029 ± 0.002 ^(b)^
	Root dry matter (g)	0.007 ± 0.005 ^(a)^	0.030 ± 0.004 ^(b)^	0.019 ± 0.004 ^(c)^
	Size of leaves (cm)	0.48 ± 0.02 ^(a)^	0.62 ± 0.02 ^(b)^	0.70 ± 0.04 ^(b)^
	Number of leaves	5.0 ± 0.5 ^(a)^	5.0 ± 0.5 ^(a)^	6.4 ± 1.0 ^(b)^
	Maximum quantum efficiency of PSII	0.70 ± 0.01 ^(a)^	0.76 ± 0.01 ^(c)^	0.73 ± 0.02 ^(b)^
	Quantum efficiency of PSII	0.29 ± 0.03 ^(a)^	0.32 ± 0.03 ^(a)^	0.34 ± 0.03 ^(a)^
	Net photosynthesis rate (AN) (µM m^−2^ s^−1^)	42 ± 9 ^(a)^	81 ± 12 ^(b)^	80 ± 10 ^(b)^
	Electron transport rate	4.8 ± 0.4 ^(a)^	9.3 ± 0.15 ^(b)^	10.9 ± 0.4 ^(c)^
	Stomatal conductance (gs)(mM m^−2^ s^−1^)	125 ± 35 ^(a)^	120 ± 34 ^(a)^	212 ± 25 ^(b)^
	Content of metals in SHOOTS (mg∙Kg^−1^)			
	As	15.4 ± 0.17 ^(a)^	18.2 ± 0.15 ^(b)^	31.1 ± 0.17 ^(c)^
	Cd	0.3 ± 0.17 ^(a)^	1.0 ± 0.02 ^(b)^	0.8 ± 0.01 ^(a)^
	Cu	5.2 ± 0.23 ^(a)^	5.9 ± 0.21 ^(a)^	11.0 ± 0.32 ^(b)^
	Zn	28.2 ± 1.1 ^(a)^	75.2 ± 0.8 ^(b)^	77.7 ± 1.7 ^(b)^
	Content of metals in ROOTS (mg∙Kg^−1^)			
	As	31.0 ± 1.4 ^(a)^	87.7 ± 2.2 ^(c)^	61.7 ± 2.2 ^(b)^
	Cd	2.0 ± 0.06 ^(a)^	3.8 ± 0.03 ^(c)^	1.4 ± 0.01 ^(a)^
	Cu	23.9 ± 0.67 ^(a)^	31.2 ± 1.11 ^(c)^	25.0 ± 0.76 ^(b)^
	Zn	149.5 ± 5.0 ^(a)^	197.1 ± 7.2 ^(c)^	94.8 ± 2.4 ^(a)^

## Data Availability

Data is contained within the article or Appendix A.

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
