# Peer review of "A Culturomics-Based Bacterial Synthetic Community for Improving Resilience towards Arsenic and Heavy Metals in the Nutraceutical Plant Mesembryanthemum crystallinum"

_ijms, 2023, doi:10.3390/ijms24087003_

Round 1
Reviewer 1 Report
Overall, the paper titled "A Culturomics-based Bacterial Synthetic Community for Improving Resilience Towards Heavy Metals in the Nutraceutical Plant Mesembryanthemum crystallinum" is well-written and presents interesting findings.
However, there are a few minor revisions needed. Firstly, the abstract should be rephrased to more clearly state the purpose and significance of the research. Also, the language of the paper could be further refined.
Additionally, it would be beneficial to discuss the limitations and future directions of the research.
Please download the attachment for further review comments.
Overall, with these minor revisions, the paper will provide valuable insights into the design of tailored biofertilizers for improving plant growth under stress conditions.

Author Response
Author's Reply to the Review Report (Reviewer 1)
Dear Reviwer # 1. Thank you very much for the evaluation of our work. We have carefully read all the comments and suggestions and we have elaborated a detailed answer to al lof them, item by item. We have highlighted the corresponding changes in the manuscript; we have used yellow color for the changes related to Reviwer # 1 and green for those related to Reviewer #2. Regarding English, a deep style correction has been performed and gramatical and/or spelling errors have been corrected. We hope that the modifications can contribute to improve and clarify teh work and that it could be now adequate for publication in International Journal of Molecular Sciences.
Overall, the paper titled "A Culturomics-based Bacterial Synthetic Community for Improving Resilience Towards Heavy Metals in the Nutraceutical Plant Mesembryanthemum crystallinum" is well-written and presents interesting findings.
However, there are a few minor revisions needed. Firstly, the abstract should be rephrased to more clearly state the purpose and significance of the research.
The abstract has been modified in order to set up the purpose of the work, as follows:
The purpose of this work was to test two different bacterial synthetic communities (SynComs) from the microbiome of Mesembryanthemum crystallinum, a moderate halophyte with cosmetic, pharmaceutical and nutraceutical applications. The SynComs were composed of specific metal resistant plant growth promoting rhizobacteria and endophytes. In addition, the possibility of modulating the accumulation of nutraceutical substances by the synergetic effect of metal stress and inoculation with selected bacteria was tested. One of the SynComs was isolated on standard tryptone soy agar (TSA), whereas the other was isolated following a culturomics approach.
Also, the language of the paper could be further refined.
Regarding English, a Deep style correction has been perfiormed and gramatical and/or spelling errors have been corrected.
Additionally, it would be beneficial to discuss the limitations and future directions of the research.
An additional paragragh has been introducen in the conclusions, as follows:
In summary, our data support the usefulness and potential of the culturomics approach to isolate specific beneficial microorganisms that stablish close relationship with plants by developing tailored culture media based on their biomass. Future studies are in progress in order to determine whether other abiotic stress situations together with inoculation with appropriate SynComs derived from the culturomics approach may booster the accumulation of metabolites with pharmaceutical/nutraceutical purposes.
Please download the attachment for further review comments.
Comments in the attached document:
Q1: tryptone soy agar (TSA)
Q2: Rephrased: The results demonstrated the effectiveness of these biofertilizers in alfalfa, improving plant growth, physiology and metal accumulation.
Q3: Paragraph moved before the introduction of the plant.
Q4: The aim a) to use a culturomics strategy for the isolation of PGPB (including bacteria from the rhizosphere and endophytes) has been deleted since it was the aim of the previous article.
Q5: The first paragraph in Results has been moved to Materials and Methods
Q6: The suggested sentences have been removed, according to the Reviewer’s comments.
Q7; In Figure 2, data a grouped by inoculation treatments. It could be also posible to group them by the absence/presence of metals.
Q8: In Figure the asterisks have been positioned in between the two columns that we wanted to compare
Q9: The error bars represent standard deviations of three determinations.
Q10: The format of Table 3 has been corrected to a triple-line table.
Q11: The discussion has been divide in subheadings, enfasizing the more innovative aspects of the work.
Q11: the number of references is in fact quite elevated. However, it has to be taken into account that different aspects are assessed in the work, namely, the interest of the plant, previous studies with it, PGPB, culturomics, plant growth and physiology, metal accumulation, metabolomics and finally the application of the SynComs in a crop plant. The diversity of techniques used and aspects assessed is the reason for the high number of references.
Overall, with these minor revisions, the paper will provide valuable insights into the design of tailored biofertilizers for improving plant growth under stress conditions.
Thank you very much for your comments and suggestions.
Responses to Reviewer # 2
Dear Reviwer # 2. Thank you very much for the review done on our work. We have carefully read all the comments and suggestions and we have elaborated an item to item detailed answer. We have highlighted the corresponding changes in the manuscript; we have used green color for the changes related to Reviwer # 2 and yellow for those related to Reviewer #1. We hope that the modifications can contribute to improve and clarify the work and that it could be now adequate for publication in International Journal of Molecular Sciences.
The topic of the study is very interesting and up-to-date. It is part of the modern approach to the impact of metals and metalloids on plants. The publication was very well prepared. From my side, there are only small remarks, which, when taken into account, contribute to the improvement of the quality of paper. My suggestions are:
- Arsenic is a metalloid and this should be indicated in the title of the paper and keywords where there is only the term heavy metals
The words metalloid or arsenic have been introduced through the whole work, in the tilte, abstract , key words and in the rest of the text and figures.
- In the Results chapter there are elements of discussion and citations of literature - this should be removed and moved to the Discussion chapter. This situation applies to: text in lines 107 - 117. 213 - 216, 250 - 252, 330 - 336, 347 - 348.
All references in the Results Section have been moved to the Discussion, together with the corresponding text (highlighted in green in the Discussion).
- Please improve the quality of figure 1 and add an explanation of the abbreviation NI in the description below the figure.
The picture was taken in the greenhouse the day that the plants were collected, we are sorry not to have a better picture.
- unify the notation in lines 445 - 447.
Corrected.
5. correct the record of mg·kg-1 in the text and in table 4
Done.
Thank you very much for your suggestions and corrections.

Reviewer 2 Report
The topic of the study is very interesting and up-to-date. It is part of the modern approach to the impact of metals and metalloids on plants. The publication was very well prepared. From my side, there are only small remarks, which, when taken into account, contribute to the improvement of the quality of paper. My suggestions are:
1. Arsenic is a metalloid and this should be indicated in the title of the paper and keywords where there is only the term heavy metals
2. In the Results chapter there are elements of discussion and citations of literature - this should be removed and moved to the Discussion chapter.
This situation applies to: text in lines 107 - 117. 213 - 216, 250 - 252, 330 - 336, 347 - 348.
3. Please improve the quality of figure 1 and add an explanation of the abbreviation NI in the description below the figure
4. unify the notation in lines 445 - 447.
5. correct the record of mg·kg-1 in the text and in table 4
Author Response
Responses to Reviewer # 2
Dear Reviwer # 2. Thank you very much for the review done on our work. We have carefully read all the comments and suggestions and we have elaborated an item to item detailed answer. We have highlighted the corresponding changes in the manuscript; we have used green color for the changes related to Reviwer # 2 and yellow for those related to Reviewer #1. We hope that the modifications can contribute to improve and clarify the work and that it could be now adequate for publication in International Journal of Molecular Sciences.
The topic of the study is very interesting and up-to-date. It is part of the modern approach to the impact of metals and metalloids on plants. The publication was very well prepared. From my side, there are only small remarks, which, when taken into account, contribute to the improvement of the quality of paper. My suggestions are:
- Arsenic is a metalloid and this should be indicated in the title of the paper and keywords where there is only the term heavy metals
The words metalloid or arsenic have been introduced through the whole work, in the tilte, abstract , key words and in the rest of the text and figures.
- In the Results chapter there are elements of discussion and citations of literature - this should be removed and moved to the Discussion chapter. This situation applies to: text in lines 107 - 117. 213 - 216, 250 - 252, 330 - 336, 347 - 348.
All references in the Results Section have been moved to the Discussion, together with the corresponding text (highlighted in green in the Discussion).
- Please improve the quality of figure 1 and add an explanation of the abbreviation NI in the description below the figure.
The picture was taken in the greenhouse the day that the plants were collected, we are sorry not to have a better picture.
- unify the notation in lines 445 - 447.
Corrected
- correct the record of mg·kg-1 in the text and in table 4
Done.
Thank you very much for your suggestions and corrections.
